# KYMASIN UP Natural Product Inhibits Osteoclastogenesis and Improves Osteoblast Activity by Modulating Src and p38 MAPK

**DOI:** 10.3390/nu14153053

**Published:** 2022-07-25

**Authors:** Laura Salvadori, Maria Laura Belladonna, Beatrice Castiglioni, Martina Paiella, Eleonora Panfili, Tommaso Manenti, Catia Ercolani, Luca Cornioli, Sara Chiappalupi, Giulia Gentili, Massimiliano Leigheb, Guglielmo Sorci, Michela Bosetti, Nicoletta Filigheddu, Francesca Riuzzi

**Affiliations:** 1Department Translational Medicine, University of Piemonte Orientale, 28100 Novara, Italy; laurasalvadori1988@gmail.com (L.S.); martina.paiella@uniupo.it (M.P.); nicoletta.filigheddu@uniupo.it (N.F.); 2Interuniversity Institute of Myology (IIM), 06132 Perugia, Italy; sara.chiappalupi@unipg.it (S.C.); giulia.gentili@studenti.unipg.it (G.G.); guglielmo.sorci@unipg.it (G.S.); 3Department Medicine and Surgery, University of Perugia, 06132 Perugia, Italy; marialaura.belladonna@unipg.it (M.L.B.); eleonora.panfili@unipg.it (E.P.); 4Department Pharmaceutical Sciences, University of Piemonte Orientale, 28100 Novara, Italy; beatrice.castiglioni@uniupo.it (B.C.); michela.bosetti@uniupo.it (M.B.); 5Laboratori Biokyma srl, 52031 Anghiari, Italy; tmanenti@biokyma.com (T.M.); catia.ercolani@biokyma.com (C.E.); lcornioli@biokyma.com (L.C.); 6Department of Health Sciences, University of Piemonte Orientale, 28100 Novara, Italy; massimiliano.leigheb@uniupo.it; 7Department of Orthopaedics and Traumatology, “Maggiore della Carità” Hospital, 28100 Novara, Italy

**Keywords:** osteoclast, RANKL, osteoblast, signaling pathways, natural product, age-related diseases

## Abstract

The imbalance in osteoblast (OB)-dependent bone formation in favor of osteoclast (OC)-dependent bone resorption is the main cause of loss of tissue mineral mass during bone remodeling leading to osteoporosis conditions. Thus, the suppression of OC activity together with the improvement in the OB activity has been proposed as an effective therapy for maintaining bone mass during aging. We tested the new dietary product, KYMASIN UP containing standardized *Withania somnifera*, *Silybum marianum* and *Trigonella foenum-graecum* herbal extracts or the single extracts in in vitro models mimicking osteoclastogenesis (i.e., RAW 264.7 cells treated with RANKL, receptor activator of nuclear factor kappa-Β ligand) and OB differentiation (i.e., C2C12 myoblasts treated with BMP2, bone morphogenetic protein 2). We found that the dietary product reduces RANKL-dependent TRAP (tartrate-resistant acid phosphatase)-positive cells (i.e., OCs) formation and TRAP activity, and down-regulates osteoclastogenic markers by reducing Src (non-receptor tyrosine kinase) and p38 MAPK (mitogen-activated protein kinase) activation. *Withania somnifera* appears as the main extract responsible for the anti-osteoclastogenic effect of the product. Moreover, KYMASIN UP maintains a physiological release of the soluble decoy receptor for RANKL, OPG (osteoprotegerin), in osteoporotic conditions and increases calcium mineralization in C2C12-derived OBs. Interestingly, KYMASIN UP induces differentiation in human primary OB-like cells derived from osteoporotic subjects. Based on our results, KYMASIN UP or *Withania somnifera*-based dietary supplements might be suggested to reverse the age-related functional decline of bone tissue by re-balancing the activity of OBs and OCs, thus improving the quality of life in the elderly and reducing social and health-care costs.

## 1. Introduction

Osteoporosis is a metabolic disease characterized by microarchitectural deterioration of bone tissue with a consequent increase in the risk of fractures, chronic pain and disability, ending in loss of independence. Osteoporosis is currently considered a major global public health issue by the World Health Organization (WHO) affecting millions of people all over the world, especially the aging population and postmenopausal women, leading to a poor quality of life and generating enormous direct health care costs [1,2,3]. Among other risks related to an unhealthy lifestyle such as unbalanced diet, stress and being sedentary, the maintenance of bone health strictly depends on the continuous and correct bone remodeling based on the coordinated interaction and balanced activity of bone reforming cells called osteoblasts (OBs), and bone-resorbing cells called osteoclasts (OCs) [3,4]. The excessive differentiation/activity of the OCs, which are multinucleated cells derived from the fusion of hematopoietic mononuclear precursors of monocytic lineage, is responsible for unbalanced bone resorption causing osteoporosis [3,4,5]. Several factors, such as chronic use of drugs, hormones, immobilization, malnutrition, and low-grade chronic inflammation, predispose to bone loss, further predisposing to fragility, morbidity and mortality [3,5]. The identification of novel anti-osteoporotic candidates with low toxicity and the ability to act in dual mechanisms, i.e., inhibition of OCs formation and increase in OBs activity, may provide an ideal strategy for treating osteoporosis. Indeed, most of the current pharmacologic or non-pharmacologic therapies have shown therapeutic effects on only one cell type, limiting the potential for clinical application [5,6,7]. The use of natural compounds in a range of diseases, including osteoporosis, is in continuous expansion due to the beneficial properties of their active metabolites [7].

Using well-characterized in vitro models mimicking muscle atrophy associated with inflammatory states, glucocorticoid treatment, or nutrient deprivation, we recently demonstrated that an herbal formulation containing *Withania somnifera*, *Silybum marianum* and *Trigonella foenum-graecum* (WST) was markedly efficacious in protecting C2C12 myotubes against reduction in the diameter and degradation of the adult isoform of myosin heavy chain (MyHC-II) [8]. In particular, in myotubes derived from sarcopenic subjects, the WST formulation rescued the typical age-related decrease in myotube size, the expression of developmental MyHC and the fusion index [8]. Based on these results, a novel dietary product called KYMASIN UP has been developed. Interestingly, the three herbal extracts combined in the KYMASIN UP and their active compounds have been reported to modulate OBs and/or OCs activities as well as bone structure [9,10,11,12,13]. The aim of the present study is to assess the effectiveness of KYMASIN UP in the differentiation process and activity of OCs and OBs by using established in vitro models consisting of the RAW 264.7 monocyte cell line conditioned with RANKL (receptor activator of nuclear factor kappa-Β ligand) protein [14,15] and C2C12 myoblasts treated with BMP2 (bone morphogenetic protein 2) [16,17], respectively. The molecular mechanisms underlying the effects of KYMASIN UP and its efficacy in human primary OBs were also investigated. We identified in KYMASIN UP a potential food supplement useful for preventing osteoporosis.

## 2. Results and Discussion

### 2.1. KYMASIN UP Suppresses the Gene Expression of Osteoclastic Markers

To investigate the potential effect of KYMASIN UP in OC formation and function, we induced differentiation of RAW 264.7 cells into OCs by treatment with RANKL (100 ng/mL) [14,15], which is the principal cytokine involved in the osteoclastic differentiation and activation during physiological bone remodeling [3,5,6,18], for 5 days in the absence or presence of the product. Firstly, different doses of KYMASIN UP (12.5–400 µg/mL) were added to RANKL-treated cultures to evaluate RAW 264.7 cell viability by MTT assay. We found that KYMASIN UP at concentrations ≥ 200 μg/mL strongly reduced cell viability (about 83% and 98% reduction at 200 and 400 μg/mL, respectively) in comparison with cells treated with RANKL alone (Appendix A). The 12.5–100 μg/mL concentration range did not alter cell viability and was considered for further evaluation.

We investigated the transcript levels of several OC markers including *Acp5* (acid phosphatase 5, tartrate resistant, TRAP), *Calcr* (calcitonin receptor), *Mmp9* (matrix metallopeptidase 9), and *Ctsk* (cathepsin K) by real-time PCR. We found that all genes were up-regulated by RANKL (fold change assigned value, 1) compared to the control (Figure 1) as expected [15], and KYMASIN UP significantly reduced the expression of these genes in a dose-dependent manner. In particular, at 100 μg/mL KYMASIN UP dramatically down-regulated the expression of *Acp5* and *Calcr*, which are highly expressed in mature OCs [19], suggesting that the product might reduce OC terminal differentiation. Moreover, 100 μg/mL KYMASIN UP completely abolished (~99% of down-regulation with respect to RANKL-treated cells) the expression of the functional OC markers, *Mmp9*, which is responsible for extracellular matrix proteolysis, and *Ctsk* which is involved in collagen degradation [20] in bone tissue (Figure 1), suggesting that KYMASIN UP might affect the functionality of OCs. These results suggest that KYMASIN UP might interfere with both OC maturation and activity by regulating skeletal osteoclastogenic genes. This is in line with previous investigations reporting that active compounds contained in *Withania somnifera* and *S. marianum* down-regulated the expression levels of *Mmp9* and *Ctsk* [11,12,21,22].

### 2.2. KYMASIN UP Inhibits RANKL-Induced Osteoclastogenesis

To evaluate if the reduction in OC markers was due to an inhibition of OC formation and/or to a direct effect on OC activity, we processed RAW 264.7 cells treated with RANKL in the absence or presence of KYMASIN UP for tartrate-resistant acid phosphatase (TRAP) staining to visualize and count purple–pink TRAP-positive multinucleated OCs (i.e., cells having three or more nuclei) (Figure 2A). Results showed that RANKL-dependent OC formation was strongly inhibited by KYMASIN UP treatment in a dose-dependent manner. Indeed, the percentage of TRAP-positive cells was significantly reduced by KYMASIN UP starting from 25 μg/mL dose and reaching the complete inhibition in the presence of 100 μg/mL of KYMASIN UP (Figure 2A). In accordance with data obtained in Figure 1, KYMASIN UP decreased TRAP enzymatic activity in a dose-dependent manner with a maximum effect at 100 μg/mL (Figure 2B). Comparing data in Figure 1 and Figure 2A,B, we observed that KYMASIN UP at the lowest dose used (i.e., 12.5 μg/mL) was able to decrease the levels of the OC functional markers, *Mmp9* and *Ctsk*, and TRAP activity but not OC numbers, showing that KYMASIN UP reduces OC activity even at a dose that does not affect OC formation. In line, all tested doses of KYMASIN UP showed a stronger effect on OC functional markers than on OC formation (Figure 1 and Figure 2A).

To further confirm the inhibition of OC differentiation by KYMASIN UP, we examined and counted multinucleated cells with F-actin rings, which are indispensable for OC adhesion to bone matrix and subsequent bone resorption [23,24], by immunofluorescence staining (Figure 2C,D). We observed that KYMASIN UP, in particular at 100 μg/mL concentration, drastically reduced the number of intact-structured F-actin rings of both small (i.e., three nuclei) and big (i.e., more than three nuclei) OCs with respect to RANKL treatment (Figure 2C,D). These results were consistent with the data obtained from TRAP staining, and collectively demonstrated that KYMASIN UP 100 μg/mL was the most efficacious concentration in counteracting RANKL-dependent OC differentiation.

The antiosteoclastogenic effect of KYMASIN UP is attributable to multiple active compounds contained in its formulation. Indeed, (i) diosgenin, a steroidal saponin present in fenugreek has been shown to suppress RANKL-dependent OC differentiation [25]; (ii) the flavonoid compound silibinin isolated from *S. marianum* inhibited OC formation by attenuating the downstream signaling cascades associated with RANKL in several in vitro models, including prostate cancer cell-induced osteoclastogenesis [12,22,26]; and, (iii) phytosterols, alkaloids and withanolides, contained in *W. somnifera*, have shown anti-osteoporotic activity [27,28].

### 2.3. KYMASIN UP Restore the RANKL/OPG Ratio in Osteoporotic Conditions

RANKL activity is inhibited by OPG (osteoprotegerin), a member of the TNF receptor superfamily, produced by numerous cell types, including OBs and bone marrow stromal cells [3,5,6,18]. OPG is a soluble decoy receptor for RANKL preventing its binding to receptor RANK and reducing the differentiation of OCs [29]. Therefore, the RANKL/OPG ratio in bone is a critical determinant in regulating the synergic activities of OBs and OCs that are at the basis of the physiological turnover in bone. Indeed, RANKL/OPG ratio is a diagnostic value for osteoporosis, with high values indicating excess of bone resorption [30]. Thus, we evaluated the release of OPG in the conditioned medium of RAW 264.7 cells treated or not with RANKL in the absence or presence of KYMASIN UP (100 μg/mL).

The results showed that RANKL nearly abolished the release of OPG from RAW 264.7 cells whereas significant amounts of OPG were still detected in the presence of KYMASIN UP, similarly to untreated cells (Figure 2E). Thus, KYMASIN UP might act in a double manner, by reducing OC formation and activity and by maintaining a physiological release of OPG also in the presence of RANKL, restoring the RANKL/OPG ratio in osteoporotic conditions. Interestingly, oral administration of silymarin or silibinin in ovariectomized mice, an experimental model of postmenopausal osteopenia, improved femoral bone mineral density and RANKL/OPG ratio, thus inhibiting femoral bone loss [21]. Moreover, treatments with withaferin A or fenugreek steroidal saponins declined RANKL/OPG ratio in OCs, by favoring the release of OPG [11] and hindering RANKL expression [31], respectively. In particular, high-dose diosgenin had a significant anti-osteoporotic effect in ovariectomized mice down-regulating the expression of RANKL and up-regulating the expression of OPG, thus lowering the RANKL/OPG ratio [32].

### 2.4. W. somnifera Is the Main Responsible for the Effects of KYMASIN UP on Osteoclasts

To test which extract(s) contained in KYMASIN UP was mostly responsible for the effects on OCs, we added KYMASIN UP (100 μg/mL) or single extracts of *W. somnifera*, *S. marianum* or *T. foenum-graecum* (same amount contained in 100 μg/mL of the dietary product, i.e., 33.3 μg/mL) to RANKL-treated RAW 264.7 cells. TRAP staining revealed that *W. somnifera* caused a highly significant reduction in OC formation (Figure 3A) and the transcription levels of all the investigated OC markers (Figure 3B) investigated in the presence of RANKL, at an extent similar to KYMASIN UP. In contrast, the presence of *S. marianum* or *T. foenum-graecum* was able to weekly reduce only *Acp5* gene expression. Thus, *W. somnifera* is the main extract responsible for the anti-osteoclastogenic effect of KYMASIN UP. In accordance with our data, withaferin A, a withanolide’s family member abundant in *W. somnifera*, has been reported to abrogate OC formation directly by reducing the expression of TRAP in vivo and in vitro experimental models [11]. Similarly, the acetyl derivative withanolide inhibited RANKL-induced osteoclastogenesis [33].

### 2.5. KYMASIN UP Does Not Affect BMP2-Dependent C2C12 Osteoblast Transdifferentiation

The treatment of C2C12 myoblasts with BMP2, playing a crucial role in skeletal development and limb formation [34], is a recognized model to induce cell differentiation into an OB lineage [16,17]. Indeed, while untreated C2C12 cells formed myotubes and expressed the myogenic marker desmin [35] after 6 days in a low serum medium, the treatment with BMP2 (300 ng/mL) reduced the presence of desmin-positive cells and converted the cell morphology into an OB-like round shape (Figure 4A). Accordingly, BMP2 markedly reduced the expressions of the myogenic transcription factor, myogenin, and the myofibrillar protein, MyHC-II [16,36] compared to untreated C2C12 cells, pointing out to myoblasts/OBs transdifferentiation (Figure 4B). KYMASIN UP did not alter the ability of BMP2 to induce OB transdifferentiation, as indicated by similar levels of desmin-positive cells and myogenic differentiation markers (Figure 4A,B). Moreover, BMP2-treated C2C12 cells increased the expression and the activity of the early marker of osteoblastic differentiation, ALP (alkaline phosphatase) [37], which was not affected by the presence of KYMASIN UP (Figure 4C and Appendix A).

We next investigated the effect of KYMASIN UP on the levels of master osteogenic factors necessary for OB maturation, i.e., *Osx* (Sp7 transcription factor, Osterix), *Col1a1* (collagen type I alpha 1 chain), *Bglap* (bone gamma-carboxyglutamate protein), and *Runx2* (RUNX family transcription factor 2) [37] in C2C12 cells at 3 and 6 days of BMP2 treatment. The results revealed that the levels of all genes were significantly increased by BMP2 (fold change assigned value, 1) compared with untreated cells (Figure 4D), as expected [16,17]. Altogether, these data suggest that KYMASIN UP does not affect the OB maturation process.

Based on reported data showing positive effects of withaferin A, silybin and silymarin on OB maturation [11,12,13], we performed experiments in BMP2-treated C2C12 cells to evaluate the contribution of the single extracts contained in KYMASIN UP in improving OB differentiation. The results revealed that the addition of *W. somnifera*, *S. marianum* or *T. foenum-graecum* alone did not modify ALP staining and activity in the presence of BMP2, similarly to KYMASIN UP (Appendix A). The discrepancy with the literature may arise from the different cell lines used and the use of total *S. marianum* and *W. somnifera* extracts instead of the single active compounds [11,12,13].

### 2.6. KYMASIN UP Improves C2C12-Derived Osteoblast Mineralization Activity

Calcein staining and relative quantification were used to evaluate the role of KYMASIN UP on OB mineralization activity. The addition of KYMASIN UP (100 μg/mL) translated into an enhanced number of mineralization nodules (Figure 5A,B) in BMP2-treated C2C12 cells, at 12 days of culture. In particular, the measure of fluorescent areas by a computer-assisted image analyzer showed values twice greater in the presence of KYMASIN UP than those detected in cells cultured with BMP2 alone (Figure 5A,B). In accordance, *W. somnifera* extract and its active compound withaferin-A, an oestrogen-like withanolide, showed positive effects in osteporotic ovariectomized animals preserving bone mineral composition [28] and reducing the proteasomal-dependent degradation of the transcription factor RunX-2 [11]. Moreover, both silymarin, the mixture of flavonolignans extracted from *S. marianum*, and its major active constituent, silibinin, promoted matrix mineralization by enhancing bone nodule formation by calcium deposit in the murine osteoblastic MC3T3-E1 cells [12,13]. Contrasting results have been reported for *T. foenum-graecum* effects on bone mineral density in several rat models. Indeed, while several reports demonstrated that *T. foenum-graecum* did not significantly affect bone mineralization [38,39], supplementation with trigonelline, the main alkaloid of *T. foenum-graecum*, ameliorated the progression of dexamethasone (Dex)-induced osteoporosis. Thus, the active metabolites contained in *W. somnifera* and *S. marianum* might be responsible for the effects of KYMASIN UP in enhancing OB mineralization capability.

### 2.7. Effects of KYMASIN UP on Human Osteoblast Activity

We tested the effects of KYMASIN UP in human pre-osteoblasts (hOBs) isolated from healthy or osteoporotic patients. Dex treatment, used as a positive differentiation control [40], modified the cell morphology of both healthy and osteoporotic hOBs increasing the amount of squared and trapezoidal cells (Figure 6A).

Moreover, Dex treatment increased ALP staining (Figure 6A) and activity (Figure 6B) on both healthy and osteoporotic hOBs. Treatment with KYMASIN UP (10 μg/mL) modified cell morphology and increased ALP staining and activity in hOBs obtained from osteoporotic patients, without affecting healthy hOBs (Figure 6A,B). In line with these data, the addition of KYMASIN UP to C2C12 cells did not affect ALP staining and activity (Appendix A), suggesting that the herbal formulation is unable to induce OB differentiation in the absence of exogenous osteogenic factors. The anti-inflammatory and anti-oxidant metabolites in the single components of KYMASIN UP might act in a stressed environment typical of osteoporotic conditions, reestablishing the correct bone deposition. Interestingly, in primary hOBs, a lower dose of KYMASIN UP (10 μg/mL) than those used above was sufficient to improve osteoporotic OB differentiation (Figure 6).

### 2.8. KYMASIN UP Affects Src and p38 MAPK Activation

To unveil the molecular mechanisms through which KYMASIN UP exerted its effects on OBs and OCs, we investigated several intracellular signaling pathways, i.e., Src (non-receptor tyrosine kinase), p38 MAPK (mitogen-activated protein kinase), ERK1/2 (extracellular signal-regulated kinases), and AKT (protein kinase B), which have been involved in RANKL-dependent osteoclastogenesis and BMP2-induced OB differentiation [41,42].

As shown in Figure 7A, RANKL promoted Src and p38 MAPK phosphorylation (activation) in RAW 264.7 cells after 5 days of treatment. These increases were significantly reduced in the presence of KYMASIN UP (Figure 7A). Also, RANKL up-regulated ERK1/2 activation without interfering with AKT phosphorylation, and KYMASIN UP did not affect ERK1/2 and AKT activation states with respect to RANKL treatment (Appendix A). Our results confirm that Src, p38 MAPK and ERK1/2 are involved in RANKL-induced OC differentiation/activation and suggest that the inhibition of KYMASIN UP on osteoclastogenesis is principally associated with the reduction in Src and p38 MAPK signaling pathways.

On the other hand, we found that BMP2 significantly activated (phosphorylated) p38 MAPK and deactivated Src, without modulating AKT and ERK1/2 signaling (Figure 7B and Appendix A). In the presence of KYMASIN UP, the phosphorylation extents of Src and p38 MAPK were lower and higher, respectively, compared to BMP2-treated cells (Figure 7B). Thus, KYMASIN UP appears able to reinforce BMP2 in activating p38 MAPK and deactivating Src, which are both events responsible for OB maturation and activity. The activation extent of ERK1/2 and AKT were not affected by treatment with BMP2 in the absence or presence of KYMASIN UP (Appendix A).

The different modulation of Src activation by KYMASIN UP is coherent with its effects on OB and OC differentiation. In particular, the anti-osteoclastogenic effect of KYMASIN UP (Figure 1) might be due to the reduction in RANKL-dependent Src activation, which has been involved in OC differentiation, actin organization, and activation of osteoclastogenic-related gene expression [43,44]. In the presence of BMP2, KYMASIN UP might favor bone mineralization (Figure 5) by further down-regulating Src phosphorylation in accordance with the known negative role of Src in OB activity [45].

p38 MAPK plays a positive role in OC proliferation, differentiation, survival, and activity [46] and in BMP2-induced C2C12 trans-differentiation [42]. These data suggest that KYMASIN UP exerts different effects on p38 MAPK depending on the cell type. KYMASIN UP reduces RANKL-dependent p38 MAPK phosphorylation in OCs thus hampering their differentiation/activity, and it enhances BMP2-dependent p38 MAPK phosphorylation in OBs, thus improving their mineralization activity.

Considering that the activation of ERK1/2 and AKT are unaffected by KYMASIN UP, we rationally conclude that KYMASIN UP effects on OBs and OCs strictly depend on Src and p38 MAPK.

## 3. Materials and Methods

### 3.1. KYMASIN UP

Based on previous results [8], a novel dietary product called KYMASIN UP was developed by Laboratori Biokyma S.r.l, Anghiari (AR, Italy) and registered to the Italian Ministry of Health (N. 128952). The product was classified according to the Directive 2002/46/CE, DM 10 August 2018, as containing a combination of *Trigonella foenum graecum* (50% seed powdered and 50% dry extract (drug/extract ratio of 4:1)), *Silybum marianum* (50% fruit powdered and 50% dry extract containing 80% of silymarin) and *Withania somnifera* (50% fruit powdered and 50% dry extract with a minimum of 2.5% concentration of withanolides). Species identification of the three officinal plants was performed by DNA barcoding technique. The single ingredients mixed in the product were provided by Laboratori Biokyma. Chemical–physical analysis (heavy metals, pesticides, pyrrolizidine and tropane alkaloids, polycyclic aromatic hydrocarbons-benzo(a)pyrene), as well as microbiological analysis (total microorganisms count, yeast and molds, ochratoxin and aflatoxin, pathogens) were performed in order to verify the quality and safety of the ingredients in accordance with UNI EN ISO 9001 quality management certification.

### 3.2. Cell Cultures

The murine macrophage cell line RAW 264.7 was obtained from the American Type Culture Collection (ATCC, Manassas, VA, USA). Cells were cultured in RPMI-1640 medium supplemented with 10% heat-inactivated Fetal Bovine Serum (FBS, Gibco), 2 mM L-glutamine (L-gln), and antibiotics (100 U/mL penicillin, 100 μg/mL streptomycin; P/S). In the experiments, RAW 264.7 cells were seeded into wells and incubated overnight (O.N.) prior to treatment. Supernatants were then replaced with a medium containing or not RANKL (100 ng/mL) to induce differentiation into OCs [14,15], in the absence or presence of different amounts of KYMASIN UP (12.5–400 μg/mL) or single components (33.3 μg/mL) for 5 days.

Murine C2C12 myoblasts, obtained from ATCC, were grown in high-glucose (4500 mg/L) Dulbecco’s Modified Eagle’s Medium (DMEM, Gibco) supplemented with 20% FBS and P/S (growth medium, GM). Differentiation into myotubes or OBs was induced by shifting myoblasts to DMEM supplemented with 5% FBS (differentiation medium, DM) for 6 days in the absence or presence of 300 ng/mL of recombinant BMP2, respectively [16,17].

Primary hOBs were obtained by enzymatic digestion from bone trabecular fragments taken at surgery and provided by the Orthopedic Institute, “Maggiore della Carità” Hospital, Novara, Italy. Written informed consent, specifying that residual material destined to be disposed of could be used for research, was signed by each participant before the biological materials were removed, in agreement with Rec (2006)4 of the Committee of Ministers Council of Europe on research on biological materials of human origin.

Samples were obtained from the femur heads or iliac crest of individuals (*n* = 3; aged 18–60 years) without diseases that could affect bone metabolism. Osteoporotic bone samples were obtained from old individuals (*n* = 3; aged 75–90 years) suffering from a low-energy traumatic hip fracture without medical conditions causing secondary osteoporosis. Cells outgrowth from the digested bone fragments appeared in culture dishes within one week, formed a confluent monolayer at 3–4 weeks, and were used in the experiments by the seventh passage after characterization for ALP expression [47]. Human cells were maintained in Iscove’s Modified Dulbecco’s Medium (IMDM, Euroclone) supplemented with 10% FBS (Hyclone GE Healthcare, Logan, UT, USA), penicillin 50 U/mL and streptomycin 15 µg/mL, and 2 mM L-gln. Human cells were treated with or without KYMASIN UP (10 µg/mL) or Dex (10 nM) as a positive differentiation control. Cell incubations were performed at 37 °C in humidified 5% CO_2_ atmosphere.

### 3.3. MTT 3-(4,5-Dimethylthiazol-2-yl)-2,5-Diphenyltetrazolium Bromide) Assay

RAW 264.7 cells (3.0 × 10^3^ cells/well) were seeded into flat-bottom 96-well plates and cultured O.N. for cell adhesion. Cells were incubated with scalar concentrations of KYMASIN UP in the presence of RANKL (100 ng/mL) for five days, being provided with a fresh medium containing stimuli on the third day of culture. Then, adherent cells were washed once and incubated for 4 h at 37 °C with 110 μL of medium containing MTT 50 μg (Sigma Aldrich, St. Louis, MO, USA). Then, 100 μL of solubilization buffer (SDS 10% in HCl 0.01 M) were added to each well, the plate was incubated overnight at 37 °C, and absorbance at 570 nm was measured using a UV/visible spectrophotometer (TECAN, Thermo Fisher Scientific, Waltham, MA, USA) to construct a graph of cell viability. The assay was performed in triplicate for each concentration.

### 3.4. TRAP Staining and Activity

RAW 264.7 cells were seeded (3.0 × 10^3^ cells/well) into flat-bottom 96-well plates for O.N. adherence, and their differentiation into OCs was obtained as previously described [15]. Briefly, cells were treated with scalar concentrations of KYMASIN UP in the presence of RANKL (100 ng/mL) for 5 days, receiving a fresh culture medium on the third day. Then, cells were stained with TRAP reagent (Kit 387-A; Sigma-Aldrich, St. Louis, MO, USA) according to the manufacturer’s protocol and acquired by a bright-field light microscope (Olympus IX51) at 10 × magnification. TRAP-positive multinucleated cells with three or more nuclei were considered OCs, and their numbers were counted in randomly selected visual fields in different areas of each well using Image J software (freely downloadable from https://imagej.nih.gov/ij/; accessed on 20 January 2022).

TRAP activity was analyzed in 30 μL of harvested supernatants by adding 170 μL of TRAP reaction buffer (Kit 387-A; Sigma-Aldrich, St. Louis, MO, USA). After 2.5-h of incubation at 37 °C, absorbance at 540 nm was measured using a UV/visible spectrophotometer (TECAN, Thermo Fisher Scientific, Waltham, MA, USA).

### 3.5. F-Actin Ring-Formation Assay

RAW 264.7 cells were seeded (3.0 × 10^3^ cells/well) on 96-well plates and treated or not with the indicated stimuli. After 5 days, the cells were fixed with 4% paraformaldehyde (PFA) for 20 min and F-actin ring-formation was evaluated by staining cells for 60 min at 37 °C with 10 µg/mL rhodamine-conjugated phalloidin (Sigma-Aldrich). Nuclei were then stained with 4′,6-diamidino-2-phenylindole (DAPI, Sigma-Aldrich). The samples were viewed by an epifluorescence microscope (Leica DMRB, Wetzlar, Germany) equipped with a digital camera.

The number of OCs in 40 µm^2^ areas was counted in 9 random fields in triplicate and reported as F-actin rings positive cells containing 3 or more than 3 nuclei.

### 3.6. Real-Time PCR

RNA extraction, reverse-transcription and real-time PCR analyses of mRNA contents were performed as previously described [48]. Calculation was performed with the specific software MXPRO-Mx 3000P (Agilent) in comparison with a standard gene (*Gapdh*). The primers used for real-time PCR analysis are reported in Table 1.

### 3.7. Western Blotting

Cells were lysed in protein extraction buffer as described in [48]. Equal amounts of total protein extract (20 to 30 µg) were resolved by SDS-PAGE (Sodium Dodecyl Sulphate-PolyAcrylamide Gel Electrophoresis) and transferred to nitrocellulose blots (Protran^TM^, 0.45 μm). Following blocking with 5% nonfat dried milk, the primary and secondary antibodies were applied as indicated in Table 2.

Analysis of OPG in the culture medium was performed as described [48]. Briefly, culture media were clarified by centrifugation, added with 1/100 volume of 2% sodium deoxycholate and subjected to precipitation with 1/10 volume of 100% trichloroacetic acid. The resultant pellets were resuspended in Laemmli buffer and titrated with 1N NaOH to obtain the normal blue color of the sample buffer, boiled for 5 min and subjected to Western blotting (WB) using the mouse monoclonal anti-OPG antibody (E-10; Santa Cruz Biotechnology, Dallas, TX, USA). The immune reactions were developed by enhanced chemiluminescence. C-DiGit Blot Scanner (LI-COR, Lincoln, NE, USA) was used for blot analysis.

### 3.8. Immunoflorescence (IF) for Desmin Expression

C2C12 cells were seeded (2.0 × 10^4^ cells/well) and cultivated on sterile glass coverslips and treated or not with indicated stimuli for 6 days. Cells were photographed at phase-contrast microscope (Olympus IX51), and then fixed with cold methanol, permeabilized using 0.1% Triton X-100 in PBS, blocked with blocking buffer containing 1% glycine (SERVA) and 3% bovine serum albumin (BSA, Sigma-Aldrich) in PBS, and incubated in a humid chamber O.N. at 4 °C with mouse monoclonal anti-desmin (Sigma-Aldrich, D-1033) primary antibody in PBS containing 3% BSA. The next day, coverslips were incubated with anti-mouse Alexa Fluor 594-conjugated antibody (Thermo Fisher Scientific) in PBS containing 3% BSA, in a light-tight humid chamber and counterstained with DAPI. Coverslips were mounted with fluorescent mounting medium containing 80% glycerol and 20% PBS and viewed in an epifluorescence microscope (Leica DMRB) equipped with a digital camera.

### 3.9. ALP Activity and Staining

C2C12 cells (2.0 × 10^4^ cells/well) and human pre-OBs (1.0 × 10^4^ cells/well) were seeded in 24-multiwell plates for O.N. adherence, and then treated or not with proper stimuli. ALP staining was performed after 6 days (C2C12) or 12 days (human pre-OBs) by using an alkaline buffer solution (100 mM NaCl, 5 mM MgCl_2_, 100 mM Trizma base pH 9.5) supplemented with 3.3 µL BCIP (5-bromo-4-chloro-3-indolyl phosphate) and 6.6 µL NBT (*p*-nitroblue tetrazolium chloride) chromogen. This solution was added to formalin-fixed cells. After 3 h of incubation at 37 °C the cytoplasm of cells containing the enzyme was stained violet/blue [49].

At the same time points, proliferation was tested using the ATP (adenosine triphosphate) quantification Kit (ViaLightTM Plus kit, Lonza, Rockland, ME, USA) whereas differentiation was quantified measuring ALP activity [49]. Briefly, culture media was removed and the cells after PBS washing were lysed with 150 µL cell lysis buffer (Tris HCl 0.05 M, SDS 0.05% pH 8) for 5 min at 50 °C. The lysates (25 µL) were supplemented with 75 µL ATP monitoring reagent. The light produced was measured by a luminometer microplate reader (VICTOR3V TM, PerkinElmer, Inc., Waltham, MA, USA) and expressed as relative luminescence units (RLUs). For ALP activity 100 µL of substrate solution (para nitrophenyl-phosphate added to Tris HCl 0.25 M pH 9.5 and 1 mM MgCl_2_) were added to 100 µL cell lysate. The mixture was incubated at 37 °C for 30 min. and absorbance was read at 405 nm in a microplate spectrophotometer. The results showed ALP activity normalizing to ATP, the index of cell number. All reagents for ALP staining and activity were from Sigma-Aldrich (Milano, Italy).

### 3.10. Mineralization Assay

In mineralization studies, C2C12 were seeded at 1.5 × 10^4^ cells/well in 24-multiwell plates and cultured in DMEM containing 5% FBS, 5 mM β-glycerophosphate, and 50 μg/mL l-ascorbic acid, and 10 nM Dex [49]. Cells were incubated or not with 300 ng/mL of recombinant BMP2, in the absence or presence of KYMASIN UP (100 µg/mL) for 12 days and the medium was replaced every three days. Mineralization was evidenced by adding to culture medium 5 µg/mL calcein (Sigma-Aldrich) O.N. Images were acquired using a Leica FLEXCAMC1 camera and mineralized nodules were analyzed using image analysis software (Qwin; Leica Microsystems, Wetzlar, Germany) in three random calcein fluorescence images in fields of 0.66 mm^2^ in each sample. Data were reported as calcein optical density = % grayscale index × area (10 − 4) in pixel.

## 4. Conclusions

In conclusion, our study demonstrates for the first time that the dietary product KYMASIN UP dramatically inhibited RANKL-induced OC formation by reducing the activity of TRAP and the expression of key genes responsible for OC activity. *W. somnifera* appears to be responsible for the inhibitory effect of KYMASIN UP, likely by down-regulating pathways under the activation of RANK receptor, i.e., Src and p38 MAPK. On the other hand, even if KYMASIN UP is unable to induce C2C12 trans-differentiation to OBs, it improves bone mineralization in vitro and rescues the maturation of human osteoporotic pre-OBs. The beneficial effect of KYMASIN UP on calcium deposition could depend on the synergic modulation of Src and p38 MAPK pathways, affected by the osteogenic factor, BMP2. These findings suggest that KYMASIN UP might be a promising candidate for the treatment of osteoporosis, limiting the excessive osteoclastogenesis and promoting OB maturation in aging conditions.

The presence of osteoporosis in elderly subjects is frequently associated with the presence of sarcopenia, defined as loss of skeletal muscle mass (muscle atrophy) and strength in a condition called osteosarcopenia [50]. In addition to mechanical linking, growing data suggest the presence of metabolic crosstalk between muscle and bone tissues by the secretion of soluble factors (myokines and osteokines, respectively) affecting the function of each other [51]. We reported that the WST herbal formulation was able to rescue the size and myogenic potential of myotubes derived from sarcopenic subjects [8].

Thus, KYMASIN UP might be used to prevent osteosarcopenia, improving the elderly quality of life and reducing social and health-care costs.

Due to the discoveries of basic research and to the ever growing collaboration between biological scientists and physicians of various specializations, the application of translational medicine and transdisciplinary in recent years has brought increasing attention to bone and muscle health through an optimal management in which an important space for nutrition is given in addition to traditional pharmacological and rehabilitative approaches [52]. Based on our promising in vitro results, further preclinical studies should be performed to evaluate the efficacy and the absorption of the dietary product in animal models of osteoporosis such as pre-geriatric and geriatric mice [53] developing age-related osteoporosis, ovariectomized mice mimicking osteoporosis in postmenopausal women or RANKL-treated mice, which rapidly lose bone tissue [54,55]. Consequently, clinical studies could consolidate the administration of KYMASIN UP even more as a tool to prevent the loss of bone mass, especially in elderly people.

## Figures and Tables

**Figure 1 nutrients-14-03053-f001:**
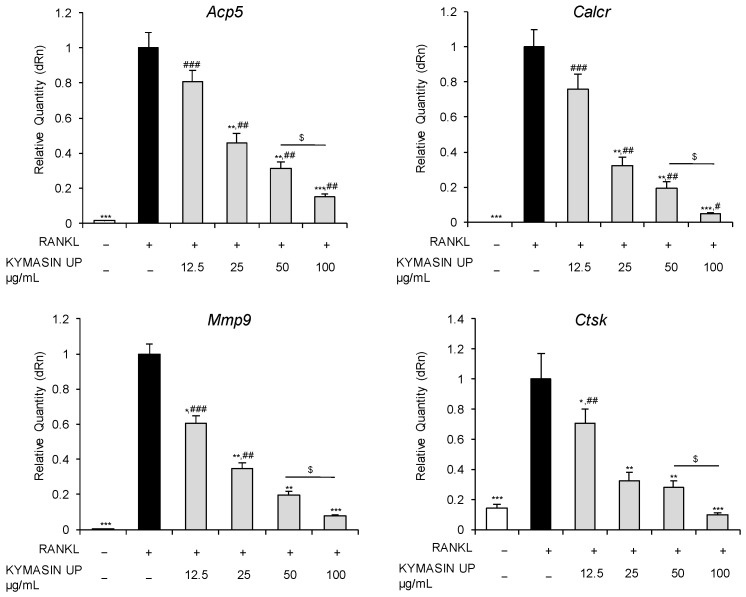
RAW 264.7 cells were treated for 5 days with RANKL (receptor activator of nuclear factor-kappa Β ligand) in the absence (black bars) or presence of KYMASIN UP (12.5–100 µg/mL) (grey bars). The untreated control was also shown (white bars). Real-time PCR analysis of the osteoclast (OC) markers, *Acp5* (acid phosphatase 5, tartrate resistant), *Calcr* (calcitonin receptor), *Mmp9* (matrix metallopeptidase 9), and *Ctsk* (cathepsin K) was performed. Gene expressions were normalized to *Gapdh*. Six independent experiments were performed. Statistical analysis was conducted using the two-tailed *t*-test. * *p* < 0.05, ** *p* < 0.01, and *** *p* < 0.001, significantly different from RANKL. ^#^ *p* < 0.05, ^##^ *p* < 0.01, and ^###^ *p* < 0.001, significantly different from untreated control. ^$^ *p* < 0.05, significantly different.

**Figure 2 nutrients-14-03053-f002:**
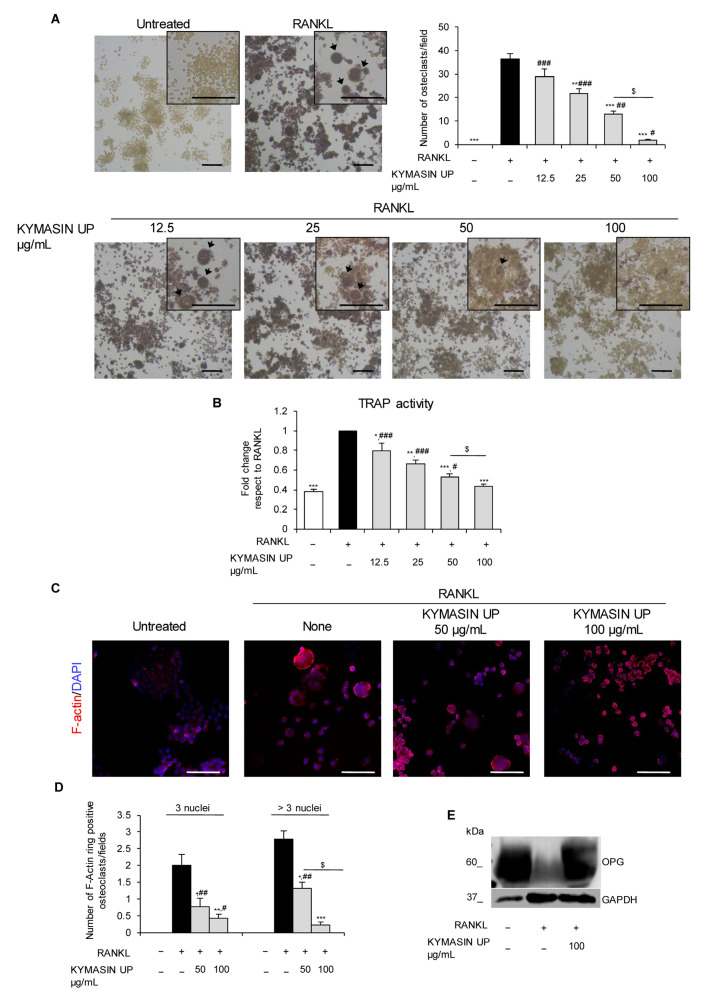
(**A**–**E**) RAW 264.7 cells were treated for 5 days with RANKL in the absence (black bars) or presence of KYMASIN UP (12.5–100 µg/mL) (grey bars). The untreated control was also shown (white bars). (**A**) TRAP (tartrate-resistant acid phosphatase) staining was performed and TRAP-positive OCs (≥3 nuclei) were counted. Representative images and high-magnification insets are reported (arrows mark OCs). (**B**) The supernatants of cells in A were collected, and TRAP activity was measured by ELISA and reported as fold change vs. RANKL treatment (black bar). (**C**) The F-actin ring (red) formation in RANKL-induced OCs was assessed using phalloidin fluorescence staining. DAPI (4′,6-diamidino-2-phenylindole, blue) was used to stain nuclei. (**D**) The number of F-actin rings positive cells containing ≥3 nuclei was determined. (**E**) Conditioned media were collected, trichloroacetic acid precipitated and subjected to Western blotting for detection of released OPG (osteoprotegerin). GAPDH relative to cell lysates is included as a loading control. Results are means ± standard error of the mean (**A**) or standard deviation (**B**,**D**). Six (**A**,**B**) or three (**C**–**E**) independent experiments were performed. Statistical analysis was conducted using the two-tailed *t*-test (**A**,**B**), and the one-way ANOVA Tukey’s test (D). * *p* < 0.05, ** *p* < 0.01, and *** *p* < 0.001 significantly different from RANKL. ^#^ *p* < 0.05, ^##^ *p* < 0.01, and ^###^
*p* < 0.001, significantly different from untreated control. ^$^
*p* < 0.05, significantly different. Scale bars (**A**,**C**), 100 μm.

**Figure 3 nutrients-14-03053-f003:**
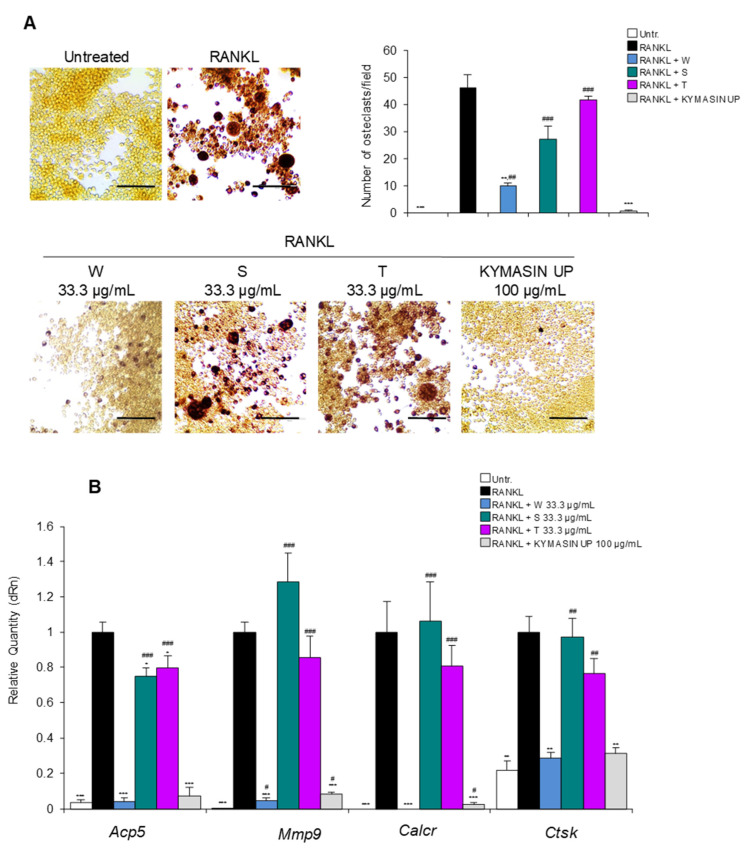
(**A**,**B**) RAW 264.7 cells were treated for 5 days with RANKL in the absence or presence of *W. somnifera* (W), *S. marianum* (S), *T. foenum-graecum* (T) or KYMASIN UP at the indicated concentrations. (**A**) TRAP staining was performed and representative images are reported. The numbers of TRAP-positive OCs (≥3 nuclei) were determined. (**B**) Real-time PCR analysis of *Acp5*, *Mmp9*, *Calcr*, and *Ctsk* was performed. Gene expressions were normalized to *Gapdh*. Results are means ± standard deviation (**A**,**B**). Three independent experiments were performed. Statistical analysis was conducted using the two-tailed *t*-test. * *p* < 0.05, ** *p* < 0.01, and *** *p* < 0.001, significantly different from RANKL. ^#^
*p* < 0.05, ^##^ *p* < 0.01, and ^###^ *p* < 0.001, significantly different from untreated control (Untr). Scale bars (**A**), 100 μm.

**Figure 4 nutrients-14-03053-f004:**
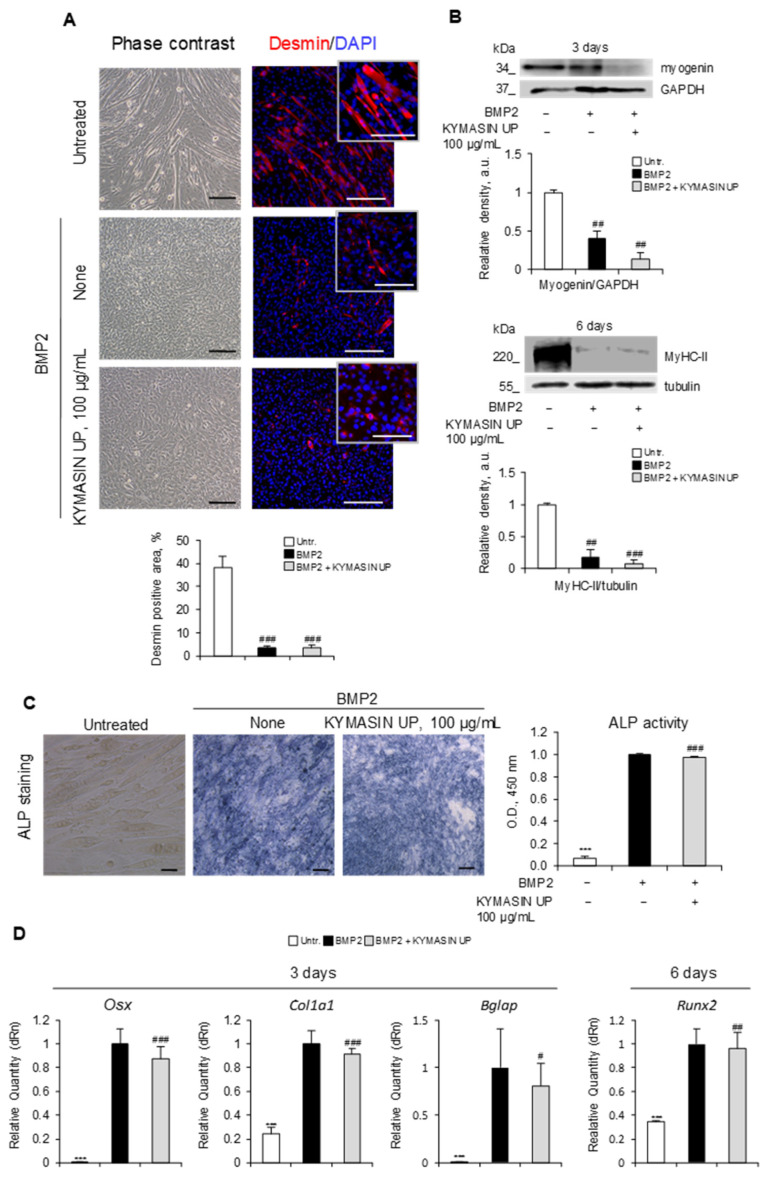
(**A**–**D**) C2C12 cells were treated with BMP2 (bone morphogenetic protein 2) with or without KYMASIN UP (100 µg/mL) for 3 or 6 days. (**A**) Phase-contrast images and immunofluorescence staining for the myogenic marker, desmin (red), were performed after 6 days of culture. DAPI (4′,6-diamidino-2-phenylindole, blue) was used to stain nuclei. The percentages of desmin-positive areas were evaluated. (**B**) The expressions of the late markers of myogenic differentiation, myogenin and MyHC-II (adult isoform II of myosin heavy chain), were analyzed by Western blotting at 3 or 6 days, respectively. Reported are the relative densities with respect to GAPDH or tubulin. (**C**) ALP (alkaline phosphatase) staining was performed, and ALP activity was measured by ELISA. (**D**) Real-time PCR analysis of the osteoblast (OB) markers, *Osx* (Sp7 transcription factor, Osterix), *Col1a1* (collagen type I alpha 1 chain), *Bglap* (bone gamma-carboxyglutamate protein), and *Runx2* (RUNX family transcription factor 2) was performed after 3 or 6 days of treatment. Gene expressions were normalized to *Gapdh*. Results are means ± standard error of the mean (**A**,**C**) or standard deviation (**B**,**D**). Three independent experiments were performed (**A**–**D**). Statistical analysis was conducted using the two-tailed *t*-test. *** *p* < 0.001, significantly different from BMP2. ^#^
*p* < 0.05, ^##^
*p* < 0.01, and ^###^
*p* < 0.001, significantly different from untreated control (Untr). Reported are representative images (**A**–**C**). Scale bars, 200 μm (**A**,**C**) or 100 μm (insets in (**A**)).

**Figure 5 nutrients-14-03053-f005:**
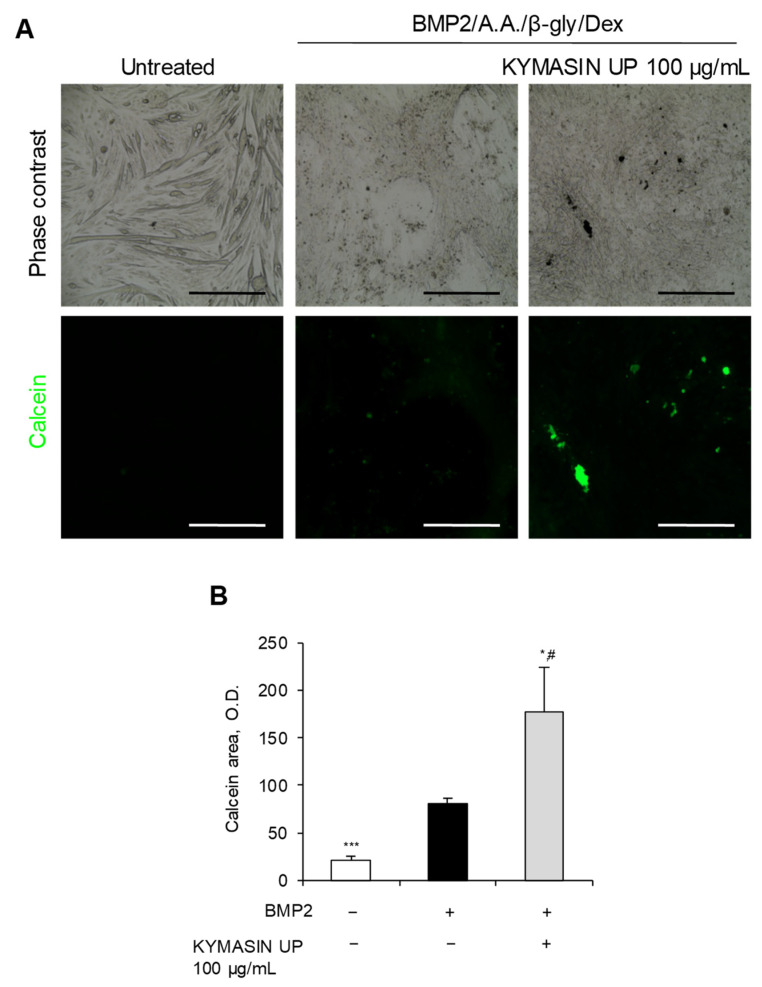
(**A**,**B**) C2C12 cells treated with BMP2 with or without KYMASIN UP (100 µg/mL) were cultured for 12 days in the presence of ascorbic acid (A.A.), β-glycerophosphate (β-gly) and dexamethasone (Dex) to induce transdifferentiation and deposition of calcium nodules. (**A**) Phase contrast images were reported. Calcium nodule deposition was observed in fluorescence after calcein (*green*) staining. (**B**) The quantification of calcein-positive nodules was performed and reported as calcein optical density. Results are means ± standard deviation. Three independent experiments were performed. Statistical analysis was conducted using the two-tailed *t*-test. * *p* < 0.05, and *** *p* < 0.001, significantly different from BMP2. ^#^
*p* < 0.05, significantly different from untreated control. Reported are representative images. Scale bars (**A**), 400 μm.

**Figure 6 nutrients-14-03053-f006:**
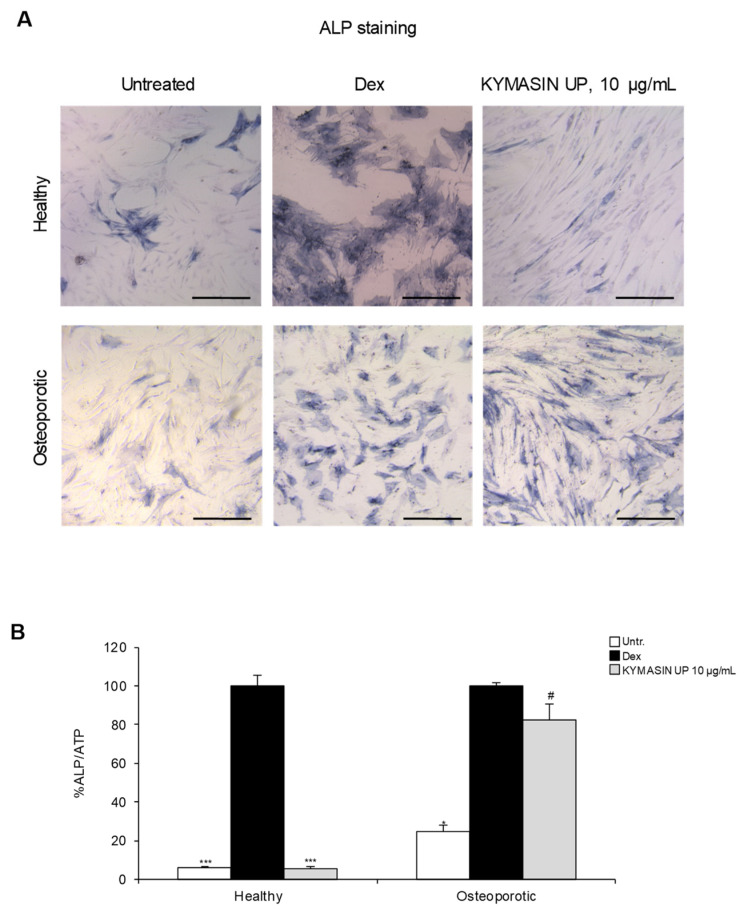
(**A**,**B**) Pre-OB cells derived from healthy and osteoporotic subjects (*n* = 3 each group) were treated for 12 days with KYMASIN UP (10 µg/mL) or with Dex as positive OB differentiation control. (**A**) Representative images of ALP staining are reported. (**B**) ALP activity was measured and normalized to ATP (adenosine triphosphate) as an index of cell numbers and reported as percentages with respect to Dex treatment. Results are means ± standard error of the mean. Statistical analysis was conducted using the two-tailed *t*-test. * *p* < 0.05, and *** *p* < 0.001, significantly different from Dex. ^#^
*p* < 0.05, significantly different from untreated control (Untr). Scale bars (**A**), 200 μm.

**Figure 7 nutrients-14-03053-f007:**
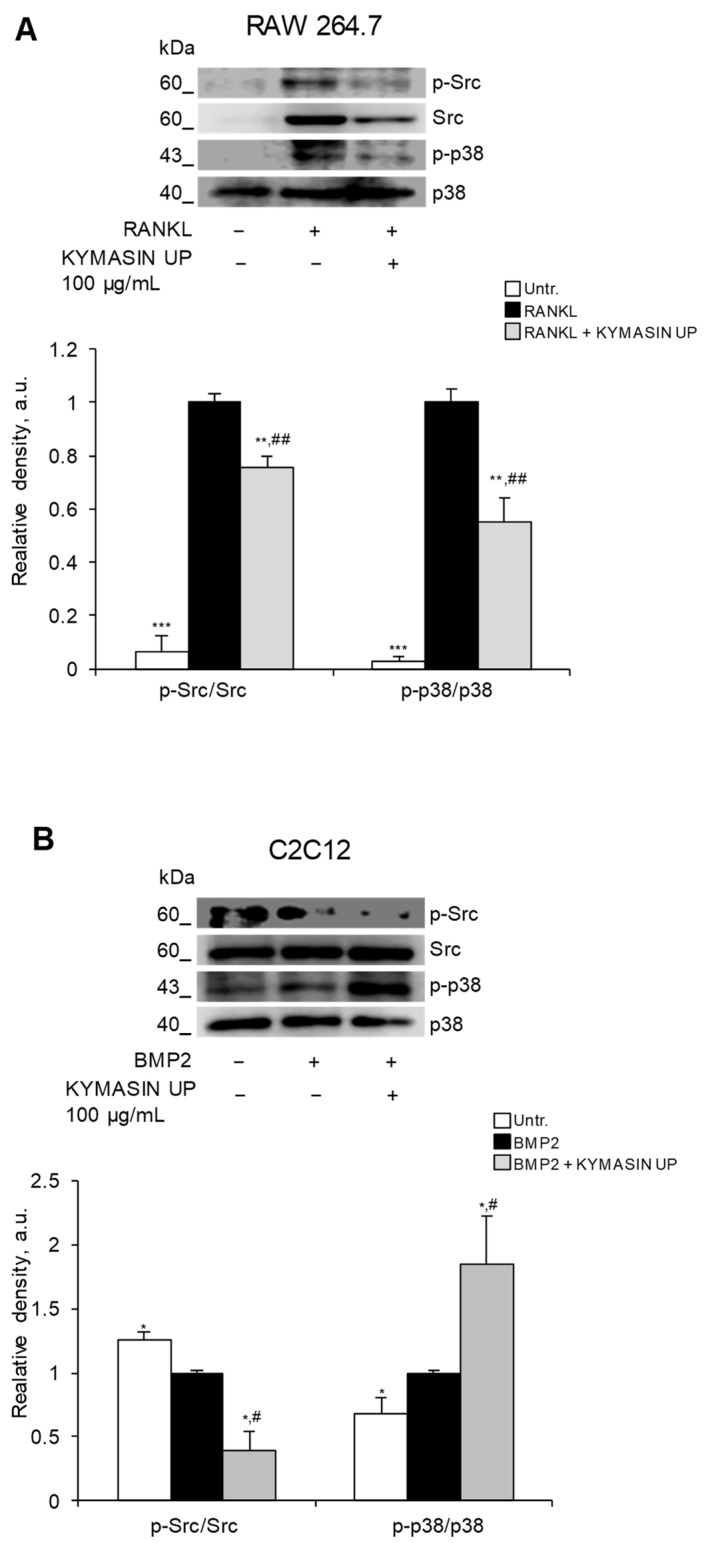
(**A**,**B**) The expressions of phosphorylated (p) Src (non-receptor tyrosine kinase) and p38 MAPK (p38 mitogen-activated protein kinase) were analyzed by Western blotting in RAW 264.7 cells treated with RANKL for 5 days (**A**) or in C2C12 cells treated with BMP2 for 6 days (**B**), in the absence or presence of KYMASIN UP (100 µg/mL). Representative images and the relative densities with respect to the total forms of Src and p38 are reported. Results are means ± standard error of the mean. Six (**A**,**B**) independent experiments were performed. Statistical analysis was conducted using the two-tailed *t*-test. * *p* < 0.05, ** *p* < 0.01, and *** *p* < 0.001, significantly different from RANKL (**A**) or BMP2 (**B**). ^#^
*p* < 0.05, and ^##^
*p* < 0.01, significantly different from untreated control (Untr).

**Table 1 nutrients-14-03053-t001:** List of primers used in real-time PCR.

Gene	Forward Primer 5′-3′	Reverse Primer 5′-3′
*Gapdh*	GCCTTCCGTGTTCCTACCC	CAGTGGGCCCTCAGATGC
*Acp5*	CTGCCTTGTCAAGAACTTGC	ACCTTTCGTTGATGTCGCAC
*Mmp9*	TAGCACAACAGCTGACTACG	ATCCTGGTCATAGTTGGCTG
*CalcR*	TCATCATCCACCTGGTTGAG	CACAGCCATGACAATCAGAG
*Ctsk*	AGAAGACTCACCAGAAGCAG	CAGGTTATGGGCAGAGATTTG
*Osx*	CCCCTTGTCGTCATGGTTACAG	AGAGAAAGCCTTTGCCCACCTA
*Col1a1*	AGCACGTCTGGTTTGGAGAG	GCTGTAGGTGAAGCGACTGT
*Bglap*	AAGCAGGAGGGCAATAAGGT	TTTGTAGGCGGTCTTCAAGC
*Runx2*	GCCGGGAATGATGAGAACTA	GGACCGTCCACTGTCACTTT

**Table 2 nutrients-14-03053-t002:** List of primary and secondary antibodies used in WB.

Primary Antibody	Molecular	Dilution	Source
	Weight (kDa)		
Mouse monoclonal anti-MyHC-II (MF20)	220	1:10,000	eBiosciences, San Diego, CA, USA
Mouse monoclonal anti-Myogenin (F5D)	34	1:1000	Santa Cruz Biotechnol., Dallas, TX, USA
Mouse monoclonal anti-α-Tubulin (DM1A)	55	1:2000	Santa Cruz Biotechnol., Dallas, TX, USA
Rabbit polyclonal anti-p38 MAPK	40	1:1000	Cell Signaling Technol., Danvers, MA, USA
Rabbit monoclonal anti-phospho-p38 MAPK	43	1.2	Cell Signaling Technol., Danvers, MA, USA
(Thr180/Tyr182) (D3F9) XP			
Rabbit polyclonal anti-Phospho-Src Family (Tyr416) #2101	60	1:1000	Cell Signaling Technol., Danvers, MA, USA
Rabbit monoclonal anti-Src (36D10)	60	1:5000	Cell Signaling Technol., Danvers, MA, USA
Rabbit polyclonal anti-phospho-p44/42 MAPK (Erk1/2) (Thr202/Tyr204)	42/44	1:2000	Cell Signaling Technol., Danvers, MA, USA
Rabbit polyclonal anti-MAP Kinase (ERK-1, ERK-2)	42/44	1:10,000	Cell Signaling Technol., Danvers, MA, USA
Rabbit monoclonal anti-phospho-Akt (Ser473) (D9E)	60	1:2000	Cell Signaling Technol., Danvers, MA, USA
Rabbit polyclonal anti-Akt (Thr308) (D25E6)	60	1:2000	Cell Signaling Technol., Danvers, MA, USA
Mouse monoclonal anti-GAPDH (6C5)	37	1:5000	Santa Cruz Biotechnol., Dallas, TX, USA
Goat anti-rabbit IgG/IgM-HRP conjugated			Sigma-Aldrich, St. Louis, MO, USA
Goat anti-mouse IgG/IgM-HRP conjugated			Sigma-Aldrich, St. Louis, MO, USA

## Data Availability

The data presented in this study are available on request from the corresponding author. The data are not publicly available in order to preserve the anonymity of the subjects involved in the study.

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
