# Peer review of "KYMASIN UP Natural Product Inhibits Osteoclastogenesis and Improves Osteoblast Activity by Modulating Src and p38 MAPK"

_nutrients, 2022, doi:10.3390/nu14153053_

Round 1
Reviewer 1 Report
The authors of this study have tested the product, KYMASIN UP, a mixture of 3 herbal extracts, on osteoclast formation and osteoblast activity using models based on cell lines. They have incorporated appropriate controls in the experiments and have derived reasonable conclusion. This manuscript can be accepted, perhaps after the following suggestions are considered:
1. Please provide the number of independent experiments performed n every figure legend.
2. In figure 1, the drug inhibits osteoclastogenesis. So, of course, the markers for osteoclasts in figure 2 would be downregulated since the numbers of osteoclasts itself are reduced as given in figure 1A. It might be better to make interchange figures 1 and 2.
3. The mention of the rat model in lines 252-258 is disruptive in that section. Please rephrase that part.
Author Response
We wish to thank the Reviewer for the criticisms, which let us to improve the quality of the paper.
We have addressed all the points raised by the Reviewer, as listed below.
Q1: Please provide the number of independent experiments performed n every figure legend.
A: We have specified the number of independent experiments performed in every figure legend.
Q2: In figure 1, the drug inhibits osteoclastogenesis. So, of course, the markers for osteoclasts in figure 2 would be downregulated since the numbers of osteoclasts itself are reduced as given in figure 1A. It might be better to make interchange figures 1 and 2.
A: We have interchanged Figure 2A and Figure 1. Former Figure 2B has been moved to Figure 2E in the revised version of the manuscript. Moreover, we have discussed the effects of KYMASIN UP on the gene levels (revised Fig. 1) vs the number of osteoclasts (revised Fig. 2A). We reported the following sentences in the revised text (lines 193-199), “Comparing data in Figure 1 and Figure 2A,B, we observed that KYMASIN UP at the lowest dose used (i.e., 12.5 μg/mL) was able to decrease the levels of the OC functional markers, Mmp9 and Ctsk, and TRAP activity but not OC numbers, showing that KYMASIN UP reduces OC activity even at a dose that does not affect OC formation. In line, all tested doses of KYMASIN UP showed a stronger effect on OC functional markers than on OC formation (Figure 1 vs Figure 2A).”
Q3: The mention of the rat model in lines 252-258 is disruptive in that section. Please rephrase that part.
A: We have rephrased the sentence as follows: “In accordance, W. somnifera extract and its active compound withaferin-A, an oestrogen-like withanolide, showed positive effects in osteporotic ovariectomized animals preserving bone mineral composition [28] and reducing the proteasomal-dependent degradation of the transcription factor RunX-2 [11]. Moreover, both silymarin, the mixture of flavonolignans extracted from S. marianum, and its major active constituent, silibinin, promoted matrix mineralization by enhancing bone nodule formation by calcium deposit in the murine osteoblastic MC3T3-E1 cells [12,13]. Contrasting results have been reported for T. foenum-graecum effects on bone mineral density in several rat models. Indeed, while several reports demonstrated that T. foenum-graecum did not significantly affect bone mineralization [38,39], supplementation with trigonelline, the main alkaloid of T. foenum-graecum, ameliorated the progression of dexamethasone (Dex)-induced osteoporosis. Thus, the active metabolites contained in W. somnifera and S. marianum might be responsible for the effects of KYMASIN UP in enhancing OB mineralization capability.” (lines 436-449).
Reviewer 2 Report
In this paper, the authors examined the effects of KYMASIN UP on osteoclasts and osteoblasts. I found it very interesting that there is a possibility that osteoporosis can be prevented from a dietary product on the market. The correction points are described below, so please revise them.
1. It would be better to discuss while discussing the active compounds in the three extracts contained in KYMASIN UP.
2. How did you decide the dose of KYMASIN UP added to the cells? Was there no cytotoxicity at this dose?
3. In conclusion, please describe future prospects (animal experiments, etc.).
Author Response
We wish to thank the Reviewer for the criticisms, which let us to improve the quality of the paper.
We have addressed all the points raised by the Reviewer, as listed below.
Q1: It would be better to discuss while discussing the active compounds in the three extracts contained in KYMASIN UP.
A: We have added discussion about the active compounds contained in KYMASIN UP along with the text (lines 149-152, 209-216, 312-319, 331-335, and 436-449)
Q2: How did you decide the dose of KYMASIN UP added to the cells? Was there no cytotoxicity at this dose?
A: We apologize for lack of clarity. As reported in the former manuscript (lines 91-96 of the revised version) “Firstly, different doses of KYMASIN UP (12.5–400 µg/mL) were added to RANKL-treated cultures to evaluate RAW 264.7 cell viability by MTT assay. We found that KYMASIN UP at concentrations ≥200 μg/mL strongly reduced cell viability (about 83% and 98% reduction at 200 and 400 μg/mL, respectively) in comparison with cells treated with RANKL alone (Figure S1). The 12.5–100 μg/mL concentration range did not alter cell viability and was considered for further evaluation.”. Then, we found 100 μg/mL as the most efficacious dose of KYMASIN UP in reducing the expression of osteoclast genes and osteoclast formation (Figures 1 and 2 of the revised manuscript) and we used this dose for the subsequent experiments. We have specified this in the revised version (lines 206-208).
Q3: In conclusion, please describe future prospects (animal experiments, etc.).
A: We thank the Reviewer for this suggestion. We have included considerations about future prospects in the Conclusions section, “Based on our promising in vitro results, further preclinical studies should be performed to evaluate the efficacy and the absorption of the dietary product in animal models of osteoporosis such as pre-geriatric and geriatric mice [53] developing age-related oste-oporosis, ovariectomized mice mimicking osteoporosis in postmenopausal women, or RANKL-treated mice, which rapidly lose bone tissue [54,55]. Consequently, clinical studies could even more consolidate the administration of KYMASIN UP as a toll to prevent the loss of bone mass, especially in elderly people.” (lines 773-779).
Round 2
Reviewer 2 Report
None.